# Is PTSD-Phenotype Associated with HPA-Axis Sensitivity? Feedback Inhibition and Other Modulating Factors of Glucocorticoid Signaling Dynamics

**DOI:** 10.3390/ijms22116050

**Published:** 2021-06-03

**Authors:** Dor Danan, Doron Todder, Joseph Zohar, Hagit Cohen

**Affiliations:** 1Anxiety and Stress Research Unit, Beer-Sheva Mental Health Center, Ministry of Health, Faculty of Health Sciences, Ben-Gurion University of the Negev, Beer-Sheva 84170, Israel; doriandanan@gmail.com (D.D.); doron.todder1@PBSH.HEALTH.GOV.IL (D.T.); 2Post-Trauma Center, Sheba Medical Center, Tel Aviv 52621, Israel; joseph.zohar@sheba.health.gov.il

**Keywords:** posttraumatic stress disorder, hypothalamus–pituitary–adrenal axis, pulsatility, animal model, corticosterone, glucocorticoid receptor, mineralocorticoid receptor, FK506-binding protein 51 (FKBP5), arginine vasopressin

## Abstract

Previously, we found that basal corticosterone pulsatility significantly impacts the vulnerability for developing post-traumatic stress disorder (PTSD). Rats that exhibited PTSD-phenotype were characterized by blunted basal corticosterone pulsatility amplitude and a blunted corticosterone response to a stressor. This study sought to identify the mechanisms underlining both the loss of pulsatility and differences in downstream responses. Serial blood samples were collected manually via jugular vein cannula at 10-min intervals to evaluate suppression of corticosterone following methylprednisolone administration. The rats were exposed to predator scent stress (PSS) after 24 h, and behavioral responses were assessed 7 days post-exposure for retrospective classification into behavioral response groups. Brains were harvested for measurements of the glucocorticoid receptor, mineralocorticoid receptor, FK506-binding protein-51 and arginine vasopressin in specific brain regions to assess changes in hypothalamus–pituitary–adrenal axis (HPA) regulating factors. Methylprednisolone produced greater suppression of corticosterone in the PTSD-phenotype group. During the suppression, the PTSD-phenotype rats showed a significantly more pronounced pulsatile activity. In addition, the PTSD-phenotype group showed distinct changes in the ventral and dorsal CA1, dentate gyrus as well as in the paraventricular nucleus and supra-optic nucleus. These results demonstrate a pre-trauma vulnerability state that is characterized by an over-reactivity of the HPA and changes in its regulating factors.

## 1. Introduction

Glucocorticoid (GC) hormones are secreted to the bloodstream by the adrenal gland and, upon a stressful event, play a role in orchestrating the physiological and behavioral reactions essential for the restoration of homeostasis after emotional or physical stress [1,2]. As such, these hormones enable the organism to prepare for, respond to, and cope with the acute demands of physical and emotional stressors [3]. Inadequate GC release following stress can lead to a delay in recovery and long-term disruptions of memory integration processes by interfering with the processing of stressful information [4,5]. One clinical example of this is post-traumatic stress disorder (PTSD) [6,7]. GCs play a major role in the development of PTSD, and lower cortisol levels in the acute aftermath of trauma were found to be a predictor for the development of subsequent PTSD symptoms [8,9,10,11,12].

Under unstressed conditions, GC hormones are secreted in a highly dynamic fashion with a characteristic circadian pattern of secretion of corticosterone. In addition to a circadian release, there is a distinct pattern of a rapid ultradian corticosterone secretion which consists of a near-hourly pulse frequency [13,14,15]. This corticosterone pulsatility is inherent within the pituitary–adrenal system and arises due to intrinsic positive feedback (activation) and negative feedback (inhibition) loops in the HPA axis as shown by in-vivo experimental work as well as mathematical modeling [14,15,16,17,18,19].

Our group has initiated a series of studies using a well-validated animal model for PTSD that examine the role of GCs in susceptibility to extreme disruption of behavior in response to stress (i.e., “PTSD-phenotype”) [20,21,22]. The initial response of the HPA axis was found to play a crucial role in producing a normative and adaptive stress response as well as in determining the long-term neuronal and hormonal imbalances underlying the behavioral symptoms of PTSD [22,23,24]. It is still unclear whether the poor corticosterone/ cortisol response to stress and the dysregulation observed in the acute aftermath of trauma represents an existing pre-trauma vulnerability trait or develops from the exposure to the trauma itself [10]. Recently, we have found that reduced basal, pre-trauma, corticosterone pulse amplitude is a pre-existing susceptibility factor for a blunted corticosterone stress response and the development of subsequent PTSD-phenotype [3]. 

The goal of this study was to identify the mechanisms that underlie both the loss of pulsatility and the differences in downstream responses. We thus examined if the PTSD phenotype is associated with HPA axis sensitivity. We examined the feedback inhibition (suppression of corticosterone following dexamethasone administration) and other modulating factors of glucocorticoid signaling dynamics such as glucocorticoid receptor (GR), mineralocorticoid receptor (MR), FK506-binding protein 51 (FKBP5), and arginine vasopressin (AVP) levels in the hippocampus.

## 2. Results

The experimental design used here is schematically depicted in Figure 1.

### 2.1. Behavioral Responses at Day 7 Post-PSS Exposure

A broad range of variations in behavioral responses was observed in PSS-exposed animals, and several subgroups were thus identified (Figure 2). Accordingly, the animals were subdivided into groups reflecting the magnitudes of responses (including EBR, PBR, and MBR) according to the CBC model. Overall, approximately 19% of the exposed animals (*n* = 4/21) fulfilled criteria for PTSD-phenotype (EBR); 38% (*n* = 8/21) for MBR; and 43% for PBR (*n* = 9/21).

### 2.2. Methylprednisolone Suppression Test (MPST)

Corticosterone concentrations were measured before (baseline) and 20 min after methylprednisolone injection, at 10 min intervals over a period of 90 min (Figure 3). At baseline conditions, serum corticosterone concentrations were not different between the groups. Methylprednisolone was administered to naïve rats. Overall, methylprednisolone significantly reduced corticosterone concentration. Post-methylprednisolone serum corticosterone levels were significantly decreased in MBR rats as compared to EBR rats, 20, 30, 40, 60, 70, and 80 min after methylprednisolone administration (Bonferroni test *p* < 0.015, *p* < 0.025, *p* < 0.03, *p* < 0.02, *p* < 0.04, and *p* < 0.05, respectively) and compared to PBR rats 60 and 70 min after methylprednisolone administration (*p* < 0.035 and *p* < 0.05, respectively) (Figure 3A) (repeated measure (RM) analysis of variance (ANOVA): Groups: F(2, 8) = 6.6, *p* < 0.007, Time: F(7, 126) = 17.2, *p* < 0.0001, Groups-Time Interaction: F(14, 126) = 2.15, *p* < 0.015). A nonlinear (exponential decay order 1 model) fitting was carried out at the six time points following corticosterone peak concentrations and indicated significantly different decay constants (*t*1) between groups, such that EBR rats exhibited a slower decay in corticosterone levels than MBR and PBR rats (for the EBR group: *t*1¼ 6.4 ± 3.1 min, MBR group: *t*1¼ 10.0 ± 3.6 min and PBR group: *t*1¼ 14.5 ± 3.2 min, Figure 3B). Taken together, individuals with PTSD-phenotype exhibited enhanced negative feedback as evidenced by greater glucocorticoid suppression following an MPST.

We also examined methylprednisolone suppression as a percentage of baseline corticosterone values (Figure 4B and Table 1). An RM-ANOVA showed that methylprednisolone produced a significantly greater suppression of plasma corticosterone in the EBR group than in the MBR or PBR groups. The percentage of suppression was calculated for the group at the beginning of suppression, at the peak of suppression and in between these two points (at 20, 40, and 70 min):

20 min post-MPST: All rats that exhibited PTSD-phenotype (4/4) demonstrated hyper-suppression (51–90%) of corticosterone in response to methylprednisolone. In contrast, most rats in the PBR and MBR behavior groups suppressed corticosterone to less than 50% of pre-methylprednisolone levels (Pearson χ^2^ = 12.35, *p* < 0.0025).

40 min post-MPST: Most rats that exhibited PTSD-phenotype (3/4) suppressed corticosterone to more than 90% of pre-methylprednisolone levels, compared with 11.11% of the PBR group (1/9) and compared with 12.5% of the MBR group (1/8) (Pearson χ^2^ = 11.23, *p* < 0.025).

70 min post-MPST: All rats that exhibited PTSD-phenotype (4/4) suppressed corticosterone to more than 90% of pre-methylprednisolone levels, compared with 55.55% of the PBR group (5/9) and compared with 12.5% of the MBR group (Pearson χ^2^ = 11.12, *p* < 0.03) at the 60 min post-MPST. 

Taken together, impaired negative feedback regulation of the HPA function was mainly observed in the EBR group.

### 2.3. Pulsatile Activity during MPST

We next assessed the characteristics of corticosterone pulses, which were identified using the Hierarchically AdaPtive algorithm [25]. Characteristics of corticosterone pulsatility are presented in Table 2. For the analysis of corticosterone pulsatility in response to methylprednisolone administration, we first divided the time post-administration into suppression (the period between 20 and 70 min post-administration) and recovery (the period between 70 and 110 min post-administration), based on the pharmacokinetics of MPST [26], so as to isolate pulsatility from the linear trend of supersession and recovery. Amplitudes were calculated individually for each rat as relative rises between the maximal and minimal absolute corticosterone values to avoid differences due to changes in absolute values. In the suppression phase, amplitudes were completely blunted in all rats in the MBR group in response to methylprednisolone administration. Significant changes were found between groups; the EBR group showed markedly more robust pulsatile activity during the suppression phase, evident by higher average amplitudes (EBR: 0.16 ± 0.1, PBR: 0.036 ± 0.02, MBR: 0); (F(2, 18) = 4.1213, *p* < 0.035). The Fisher LSD post hoc test confirmed a significantly more prominent pulsatile activity in the EBR group as compared with the MBR group (*p* < 0.015) and the PBR group (*p* < 0.04); differences between the EBR and MBR groups were also significant for the Bonferroni post hoc test (*p* < 0.035). In the recovery phase, no significant changes were found between groups though a trend towards higher amplitudes in the EBR group was still apparent (EBR: 0.64 ± 0.15, PBR: 0.53 ± 0.13, MBR: 0.24 ± 0.09) (F(2, 18) = 2.5073, *p* = 0.11). A technical limitation in the duration of sampling made frequency analysis futile.

### 2.4. Brain Immunohistochemistry

All rats in Experiment 2 were decapitated one day following behavioral assessments (Figure 1), and brains from six unexposed controls were also harvested for comparison.

### 2.5. Brain GR-ir Cells 8 Days Post-PSS Exposure

As the negative feedback loop of the HPA axis is mediated via GR and MR, we assessed the dynamic response of GR and MR to PSS in the dorsal and ventral hippocampus by quantified MR and GR nuclear translocation patterns. Looking at GR-ir cells in the hippocampal CA1 subregion (Figure 5A), two-way ANOVA revealed a significant effect on groups (F(3, 52) = 5.06, *p* < 0.004), a significant effect on the hippocampal axis (F(1, 52) = 198, *p* < 0.0001) and on the hippocampal-axis interaction (F(3, 52) = 9.9, *p* < 0.0001). Bonferroni post hoc tests confirmed significantly higher GR-positive cells in the nuclear layer in the dorsal CA1 areas in EBR as compared with unexposed control (*p* < 0.002) and MBR (*p* < 0.04) rats. In contrast, significantly lower GR-positive cells were found in the ventral CA1 areas in EBR compared with unexposed control (*p* < 0.05) and PBR (*p* < 0.007) rats. In all groups, the expression of GR-positive cells in the nucleus was significantly higher in the dorsal part of the CA1 than in the ventral part (CONT: *p* < 0.002, EBR: *p* < 0.0001, PBR: *p* < 0.0001 and MBR: *p* < 0.0001). In the DG (Figure 5B), two-way ANOVA revealed a significant effect on the hippocampal axis (F(1,52) = 35.3, *p* < 0.0001) and a group–axis interaction effect (F(3, 52) = 15.85, *p* < 0.0001). No effects were observed for groups, but there was a clear trend showing higher GR-positive cells in the PSS-exposed groups (F(3, 52) = 2.6, *p* = 0.059). Bonferroni post hoc tests confirmed significantly higher GR-positive nuclear translocation cells in the dorsal DG areas in EBR rats compared with unexposed controls (*p* < 0.003). Significantly lower GR-positive nuclear translocation cells were found in the ventral DG areas in EBR compared with unexposed control (*p* < 0.0004), PBR (*p* < 0.0001), and MBR (*p* < 0.009) rats. In the EBR group, the expression of GR-positive cells in the nucleus was significantly higher in the dorsal part of the DG than in the ventral part (*p* < 0.0001), this was not true for other groups. There were significant differences in GR levels among the groups in the PVN (Figure 5C) (One-way ANOVA: F(3, 26) = 7.8, *p* < 0.00075). Significantly lower GR-positive cells were found in the PVN area in the EBR group than in the unexposed control (*p* < 0.02) and MBR (*p* < 0.0025) rats. PBR rats also exhibited significantly lower GR-positive cells compared to the MBR group (*p* < 0.015).

### 2.6. Brain MR-ir Cells 8 Days Post-PSS Exposure

As noted above, hippocampal MR is also thought to play a principal role in the negative feedback effects of glucocorticoids, primarily on the basal HPA tone at both the trough and peak of the diurnal cycle [27]. In the CA1 subregion (Figure 6A), two-way ANOVA revealed a significant effect on the hippocampus axis (F(1, 52) = 6.0, *p* < 0.02), and a groups–axis interaction effect (F(3, 52) = 16.4, *p* < 0.0001). No effects were observed for groups. Bonferroni post hoc tests confirmed significantly higher MR-positive cells in the nuclear layer of the dorsal CA1 areas in EBR compared with unexposed control (*p* < 0.03) and MBR (*p* < 0.00015) rats. In contrast, significantly lower MR-positive cells were found in the ventral CA1 areas in EBR compared with unexposed control (*p* < 0.045) and MBR (*p* < 0.0025) rats. In addition, in EBR and MBR rats, the expression of MR-positive cells was region-specific: EBR rats exhibited significantly higher MR-positive cells in the dorsal hippocampus compared to the ventral part (*p* < 0.05), whereas the MBR rats exhibited significantly higher MR-positive cells in the ventral hippocampus compared to the dorsal part (*p* < 0.0001).

In the DG (Figure 6B), a two-way ANOVA revealed a significant main effect on groups (F(3, 52) = 2.96, *p* < 0.045), a significant effect on the hippocampus axis (F(1, 52) = 13.0, *p* < 0.0007) and a significant group–axis interaction (F(3, 52) = 27.2, *p* < 0.0001) on the nuclear translocation profile of MR protein. Bonferroni post hoc tests confirmed significantly higher MR-positive cells in the dorsal DG areas in EBR compared with unexposed control (*p* < 0.0007), PBR (*p* < 0.02), and MBR (*p* < 0.0015) rats. In contrast, significantly lower MR-positive cells were found in the ventral DG areas in EBR compared with unexposed control (*p* < 0.0001) and MBR (*p* < 0.0001) rats. In addition, in EBR and MBR rats, the expression of MR-positive cells was region-specific: EBR rats exhibited significantly higher MR-positive cells in the dorsal hippocampus compared to the ventral part (*p* < 0.0001), whereas the MBR rats exhibited significantly higher MR-positive cells in the ventral hippocampus compared to the dorsal part (*p* < 0.0001). In the PVN (Figure 6C), no significant differences were found between exposed groups, however, all exposed groups exhibited significantly lower GR translocation compared to unexposed controls (EBR: *p* < 0.0085, PBR: *p* < 0.008 and MBR: *p* < 0.015) (One-way ANOVA: F(3, 26) = 6.8, *p* < 0.0015).

### 2.7. Brain FKBP5 Levels 8 Days Post-PSS Exposure

The FK506-binding protein-51 (FKBP5; *fkbp5* gene) regulates HPA negative feedback by preventing nuclear translocation of the GR complex (100,352). Therefore, we examined the protein levels of FKBP5 in the cytosolic fraction of ventral, dorsal hippocampus, and PVN. In the CA1 (Figure 7A) and DG (Figure 7B) subregions, two-way ANOVA revealed a significant effect for hippocampus axis [F(1, 52) = 8.5, *p* < 0.0055 and F(1, 52) = 26.8, *p* < 0.0001, respectively], and a group–axis interaction effect [F(3, 52) = 6.2, *p* < 0.0015 and F(3, 52) = 5.8, *p* < 0.002, respectively]. No effects were observed for groups. Bonferroni post hoc tests confirmed significantly lower FKBP5-positive cells in the dorsal CA1 and DG subregions in EBR compared with unexposed control (*p* < 0.05 and *p* < 0.0015, respectively), MBR (*p* < 0.01 and *p* < 0.04, respectively) and PBR (*p* < 0.04, only in the DG) rats. Furthermore, significantly higher FKBP5-positive cells were found in the ventral CA1 subregion in EBR rats compared with unexposed controls (*p* < 0.05). In addition, significantly lower FKBP5-positive cells were found in EBR rats in the dorsal subregions compared with ventral parts (*p* < 0.04 and *p* < 0.0001, respectively). In the PVN (Figure 7C), EBR rats exhibited significantly higher FKBP5-ir cells compared to controls (*p* < 0.0025) (One-way ANOVA: F(3,26) = 5.7, *p* < 0.004).

### 2.8. Brain AVP Levels 8 Days Post-PSS Exposure

Since AVP has long been recognized as a neuropeptide and involved in both the modulation of HPA axis activity [28] and negative feedback sensitization [29], AVP levels in hypothalamic neurons were evaluated using quantitative immunocytochemistry. Significant differences in AVP overall fibers were noted between groups in the PVN [F(3, 25) = 24.4, *p* < 0.0001]. Bonferroni post hoc tests confirmed significantly higher AVP-positive fibers (Figure 8A) in the PVN in EBR rats compared with unexposed control (*p* < 0.0001), MBR (*p* < 0.0001) and PBR (*p* < 0.0001) rats. Significant differences in AVP levels were noted between groups in the SON (Figure 8B) [F(3, 26) = 16.2, *p* < 0.0001]. Bonferroni post hoc tests confirmed significantly higher AVP- positive cells in the SON in EBR rats compared with unexposed controls (*p* < 0.0001), PBR (*p* < 0.0001) and MBR (*p* < 0.0001) rats. We next quantified magnocellular and parvocellular cells in the PVN separately (See methods). Significant differences in AVP in magnocellular cells were noted between groups in the PVN (F(3, 26) = 7.6203, *p* = 0.00082). Bonferroni post hoc tests confirmed significantly higher AVP-positive magnocellular cells (Figure 8C) in the PVN in EBR rats compared with unexposed control (*p* < 0.025), MBR (*p* < 0.0001) and PBR (*p* < 0.025) rats. Lesser, but still significant, differences in AVP were also found in parvocellular cells between groups in the PVN [F(3, 26) = 3.0668, *p* = 0.04556]. Bonferroni post hoc tests confirmed significantly higher AVP-positive parvocellular cells (Figure 8D) in the PVN in EBR rats compared with the MBR (*p* < 0.045) rats. These findings indicate alterations in the hypothalamic expression of AVP only in PTSD-phenotype rats in the SON and in the PVN in both the parvocellular and magnocellular neurosecretory systems.

## 3. Discussion

The mechanisms that render an individual susceptible to PTSD are complex, largely unknown, and are composed of a dynamic interplay that chiefly involves the HPA axis and its related hormones. To assess our hypothesis on whether the blunting of basal ultradian amplitudes is a pre-existing vulnerability trait caused by an over-suppression/over-reactivity of the HPA axis, we conducted an MPST test a day prior to PSS. Retrospective analysis of dynamic blood sampling of corticosterone concentrations (Table 1, Figure 3 and Figure 4) showed that all rats who displayed extreme behavioral disruption (4/4, PTSD-phenotype) and more than half (5/9, 55.5%) of those displaying significant behavioral disruption (distinct from both the extreme and the minimal response patterns)—i.e., partial responders (PBR)—had an enhanced suppression of corticosterone in response to the MPST at baseline, i.e., prior to stress-exposure. In contrast, only one individual (1/8, 12.5%) amongst those with minimally disrupted behavior had responded to methylprednisolone in this manner. These results indicate that hyper-suppression of the HPA axis in response to methylprednisolone is indicative, in part, of a pre-trauma vulnerability factor, which increases the individual’s likelihood of developing full or partial PTSD-like behavior (EBR, PBR) following stress exposure. Enhanced negative feedback of the HPA axis at baseline in rats subsequently found to exhibit PTSD-phenotype supports the previous findings that over-regulation of negative and, possibly, positive feedback mechanisms might be responsible for blunted basal corticosterone amplitudes, significantly reduced corticosterone response to the stressor, and, hence, PTSD susceptibility [3]. The dysregulation in glucocorticoid signaling (sensitization of the HPA axis and enhanced suppression of cortisol following the methylprednisolone) before the stress exposure may represent vulnerability factors for the development of PTSD [10]. This, however, explains our results only partly as not all rats who exhibited hyper-suppression subsequently developed and an extreme disruption of behavior. 

This dysregulation can theoretically be the result of faults in several pathways. To obtain a more in-depth resolution of the real-time regulation of the HPA axis occurring during the MPST, we assessed the pulsatile patterns of corticosterone during the MPST obtained by serial blood sampling. We looked at the suppression and recovery phases separately. All rats in the MBR group (8/8) presented a complete blunting of pulsatile patterns during the suppression phase, i.e., no amplitudes were detected. This was not true in the PBR group where fewer than half of the rats presented a pulse during the suppression phase (4/9 44%); these pulses were presented as “escape pulses,” with quick rises of short duration. In the EBR group, pulsatile activity was seen in all but one of the rats (3/4 75%). Average amplitudes of the EBR group were also higher in size compared to those of the PBR group (Table 2). During the recovery phase, pulsatile activity returned in all rats of the EBR and PBR groups and remained completely blunted in only 2 out of the 8 rats in the MBR group (25%). Even though the same trend of higher average amplitudes was still evident in the recovery phase, results were non-significant, which, together with the apparent return of pulses, is indicative of the resolution of the blunting of amplitudes in the MBR and PBR groups. It is important to note that due to technical difficulties, the duration of serial blood measures was not prolonged enough to demonstrate the frequency of pulses or the full return to baseline amplitudes. Even though the frequency was not significant in baseline or in post-stress conditions in our previous study, this remains a serious limitation of this study.

These results are intriguing. Taken together with our previous results, a clearer picture of the over-reactivity of the HPA axis as a contributing factor to the development of PTSD is starting to emerge. Although we were not able to test corticosterone levels at the onset of PSS, prominent feedback inhibition of corticosterone release is still evident in the PTSD-phenotype rats by the pre-trauma MPST response, although only partly. In addition, there is an implication of a more prominent forward-driving regulation of corticosterone release in rats later exhibiting a PTSD phenotype, evident by pulsatile activity during MPST and by the blunting of baseline amplitudes without a change in the mean baseline in our first experiment.

The dysregulation of the HPA axis is extensively studied in PTSD research. In a recent review [30], the most common HPA dysregulation finding in PTSD patients was an enhanced cortisol suppression following low-dose dexamethasone treatment. This led to the assumption of enhanced sensitivity of the HPA axis to glucocorticoid activation [31,32,33]. This assumption in PTSD patients indicated an increased sensitivity to GR (either by quantity or quality), and indeed, early studies, outside of the HPA axis, have suggested that the GR expression in peripheral blood mononuclear cells (PBMCs) or lymphocytes was greater in PTSD patients compared to control subjects. This appeared to conform with the observed hyper-suppression of the HPA axis to GC administration [34,35]. More recent studies, moreover, are in agreement with these findings, as they found either a reduced GR binding capacity [36,37,38,39] or no difference between PTSD patients and controls [40,41,42]. Furthermore, a recent cohort study of PTSD patients found heightened negative feedback of the HPA axis to dexamethasone but reduced GR lymphocyte expression [38,43]. These findings suggest that, though the picture of increased HPA reactivity is prominent in PTSD patients, the cause is more complex than an isolated GR abnormality. Another important question, which is yet to be answered, is whether this intricate dysregulation is a PTSD marker or a pre-trauma trait that constitutes a vulnerability factor [44]. Regarding this last question, a large prospective study by van Zuiden et al. in pre-deployment soldiers found several components of GR signaling, including greater GR numbers in PBMCs and lower FKBP5 mRNA levels, to be vulnerability factors for soldiers developing greater PTSD symptoms post-deployment in combat zones. This was significant even after controlling for childhood trauma experiences [45]. Another touching about PTSD marker versus a pre-trauma vulnerability trait is a study conducted on police academy officers. After officers were exposed to a short video of other police officers exposed to real-life trauma, salivary cortisol was measured. Officers who followed trajectories of resilience and recovery over 4 years mounted significant increases in cortisol in response to the experimental stressor, while those following a trajectory of chronically increasing distress had no significant cortisol change in response to the challenge [46]. Another study suggesting dysregulation of HPA as a pre-trauma vulnerability has found that lower salivary cortisol stress reactivity in pre-deployment soldiers was predictive of a greater increase in PTSD symptomatology in soldiers who had experienced a new onset of traumatic events [47].

Our results strongly support that the HPA-dysregulation, found in PTSD, is a pre-trauma vulnerability trait. Furthermore, it points to a dysregulation that is indeed feedback-dependent and composed of both a negative inhibition of the axis and a positive forward drive.

The positive drive of the HPA axis arising from the PVN is usually thought of as a tonic drive that is propagated by inputs from the SCN and other regions and can be either augmented by stress or blunted by feedback inhibition [48,49,50,51,52,53]. In addition, most models of HPA-regulation focus purely on negative feedback elements of the HPA [49] and rely mainly on GR for feedback regulation [48]. While a feedback inhibition of the HPA axis is important for homeostasis, it is also important to maintain cortisol concentrations above a critical threshold since low cortisol may result in a number of pathologies [54,55]. A model first suggested in a paper by Avital et al. [56] and again by Peters et al. [57] points to the importance of both a positive and negative feedback regulation of the HPA axis and implicates the binding of corticosterone to the high-affinity MR as a forward modulator of the HPA axis. Even though this model is supported by some mathematical and clinical data [49,58], it is contradicted in other studies [59]. It is, however, possible that the positive forward drive of the HPA might arise from the binding of corticosterone to the MR in the PVN itself or in “higher” regions that provide excitatory inputs to the PVN [60].

We next analyzed the harvested brains to assess the consequences of PSS exposure on GR and MR signaling in the hippocampus and hypothalamic PVN [61]. We found that the PTSD-phenotype group exhibited long-term regionally distinct changes in the expression of both MR and GR in the hippocampus and in the hypothalamic nucleus in response to PSS. In the dorsal hippocampus, GR and MR were significantly increased in the nuclear compartment in PTSD-phenotype rats in compression to control and MBR, whereas the ventral hippocampus presented reduced GR and MR translocation., In the PTSD-phenotype, there were overall more MRs and GRs present in the dorsal hippocampus in compression to MBR and control, while in the ventral hippocampus, there was an overall reduction in both MR and GR. A region-specific comparison revealed that in the EBR group, the level of GRs in the ventral DG was significantly lower than that of the dorsal DG; this was not true for the other groups, showing no significant differences in GR expression between these areas. Even more interesting is the regionally distinct differences in MR expression, while the level of MRs was significantly lower in the dorsal hippocampus in comparison to the ventral hippocampus in the control group; these differences were more prominent in the MBR (‘resilient’) group, and furthermore, these changes were significant and opposite in the EBR group, showing higher levels of MR in the dorsal hippocampus compared with the ventral hippocampus, while the PBR group showed no significant differences between areas. In the PVN, however, the EBR group exhibited a significant reduction in GR levels in comparison with the control group but also in comparison with MBR, whereas MR is equally reduced in all the exposed groups regardless of the behavioral responses (expressed phenotype). Because the behavioral phenotype is the only difference between exposed groups, it seems that while GRs in the PVN plays a role in the behavioral disruption, MRs do not seem to be related to the expression of the pathological phenotype. Therefore, the PTSD-phenotype group exhibited a different distribution of these receptors along the longitudinal axis of the hippocampus and in the PVN; these differences may affect functions such as memory storage processes, fear response, and HPA axis response. This was observed by Chen and Etkin [62] in PTSD patients compared with non-traumatized controls. 

It has been suggested that the dorsal hippocampus and ventral hippocampus play distinct functional roles; this segregation of functions along the axis of the hippocampus appears at several levels of the local network organization including principal cell properties [63], synaptic transmission and neurotransmitter receptors [64], and synaptic plasticity [65,66,67]. The balance between MR and GR mediated actions is thought to determine the stress responsiveness of an individual. Disturbance of this balance can dysregulate the stress system and enhance vulnerability to stress-related disorders [68,69]. Furthermore, MR activation not only rapidly promotes memory retrieval but also mediates, with a short delay, the effect of corticosterone on selective attention, pro-motes reactivity to novel situations, and encodes the experience for memory storage both in rodents and in humans [70,71,72,73].

To interpret our results, we first focused on the hippocampal areas that regulate the PVN and HPA axis to “trace back” the fault in regulatory mechanisms of the HPA axis activity. The ventral hippocampus, which corresponds to the anterior part of hippocampus in humans [74], is involved in stress-related responses such as anxiety, fear, and defensive behavior, and also exerts feedback control over the HPA-axis [75,76,77,78,79,80,81]. The hippocampal regulation of the HPA axis by GC is known to be inhibitory through GABAergic neurons [82,83,84,85]. Studies in rats have indicated that antagonists of MR administered via intra-hippocampal administration (i.e., induced deactivation of MR activity in the hippocampus) have led to AM and PM hyperactivity of the HPA axis, both during basal condition and in response to stress [86,87]. These findings correlate with studies in humans showing that higher doses of the MR antagonist – spironolactone – also increased basal and stress-induced cortisol secretion [73]. These effects might be mitigated through a non-genomic membrane MR [88,89,90]. It was also found that the administration of a GR antagonist rapidly attenuated the initial HPA response to stress and only later interfered with the HPA feed-back regulation, causing stress-induced corticosteroid levels to remain high for a prolonged period of time [86,87,91]. This, in conjunction with our findings, suggests that lower activity of MR and GR in the ventral hippocampus might cause the observed changes of over-activity during a suppression test, the blunted basal amplitude, and the faults in the mounting of corticosterone in response to stress, which, subsequently, led to extreme behavioral disruption. 

While it is tempting to leave it at that, this would be an over-simplification as there are several other factors to keep in mind. First, while the differences in HPA regulation were seen prior to trauma, the differences in GR and MR expression seen in the PTSD-phenotype group were measured 8 days post trauma. It is well documented that aversive events can lead to changes in receptors expression [92,93,94,95,96,97,98]. One study [99] showed that pre-trauma high levels of GR are a vulnerability factor for PTSD, though the study was done on peripheral blood leukocytes. Furthermore, while changes in the ventral hippocampus may explain part of the picture, there are many other factors involved. Although it was alluded to in a study by Malivoire et al. [100] that the “anterior hippocampus is a potentially superior biomarker compared to the posterior hippocampus”, regarding PTSD symptomology, the dorsal hippocampus, which corresponds to the posterior part of hippocampus in humans [74], is known to play a crucial role in learning, long-term memory storage, and recollection memory, through its connection with the default-mode network [75,76,77,78,79,80,81] and has been linked to PTSD both in humans and in rodents [62,101,102]. Elevated levels of GR in the dorsal hippocampus after a traumatic event have been demonstrated in the dorsal hippocampus in numerous studies [92,93,94,95,96], including our group [97]. The increase in GR expression in dorsal hippocampus seems to be an effect of the traumatic event and might play a role in ameliorating the negative effect of glucocorticoid excess on hippocampal plasticity, learning, and memory [93,94,95]. These changes in the expression of GR and MR along the hippocampal axis in the PTSD-phenotype group might modulate the sensitivity of the hippocampus to corticosteroids, thereby leading to differences in brain connectivity and modifying the entire network, which might be key in determining susceptibility to stressors. 

Our findings of reduced MR and GR in the PVN support previous findings by Louvart et al. [98], and may also be a consequence of the traumatic event. It is worth noting that there are other mechanisms other than the quantity of receptors that affect their level of activation. 

One such mechanism, contributing to receptors of glucocorticoid’s sensitivity, is a large chaperone protein complex, FKBP5. When FKBP5 is bound to the GR via HSP90, the receptor has a lower affinity for its ligand and is retained in the cytoplasm [103,104]. While less studied, FKBP5 effects on MR are similar, i.e., a reduction of its affinity and activity [105,106]. We found that the PTSD-phenotype group exhibited different distributions of FKBP5 along the longitudinal axis of hippocampus and in the PVN. In the dorsal hippocampus, FKBP5 was significantly decreased in PTSD-phenotype rats, whereas in the ventral hippocampus and PVN areas, we found increased expression of FKBP5, which may inhibit GR translocation to the nuclear compartment. Notably, differences of FKBP5 in the PTSD-phenotype group were inversely correlated with GR and MR expression, suggesting reduced activity of these receptors in the ventral DG and CA1 areas, and in the PVN, and increased activity in the dorsal DG and CA1 areas. A study in mice has shown that genetic over-expression of FKBP5, in combination with early life stress, significantly exacerbated anxiety behavior during stress exposure compared to controls, however, mice who were ex-posed to early life stress showed significantly lower levels of FKBP5 in the dorsal hippocampus compared to same-mutated mice who did not experience early life stress [107]. This might suggest that lower levels of FKBP5 in the dorsal hippocampus might be involved in susceptibility to stress. However, no changes were seen in ventral hippocampus and a different study found that overexpression of FKBP5 in the dorsal hippocampus did not have an effect on anxiety-like behaviors [108]. However, Changes in the PTSD behavioral group might lead to an impaired negative feedback mechanism and/or to glucocorticoid resistance [104]. Yet, genome wide association (GWA) studies have failed to identify any significant associations between HPA axis genes and PTSD symptoms or severity [109,110], suggesting an involvement of environmental factors with the epigenome. Among some genes related to HPA-axis, the FKBP5 gene has shown the strongest association with PTSD symptoms [111,112,113,114]. A recent meta-analysis reported that single nucleotide polymorphisms (SNPs) in FKBP5 and NR3C1 genes (rs258747 in NR3C1 and rs9296158 in FKBP5) were associated with PTSD [115]. Recently, Zhang and colleagues [116] investigated the genetic architectures of the HPA axis in relation to the stress response. They explored the relationship between HPA axis gene polymorphisms and the stress- and fear-related disorder PTSD. The authors evaluated PTSD symptoms in 1132 Chinese earthquake survivors. In the next step, the author’s genotyped SNPs of three HPA axis genes, FKBP5, CRHR1 and CRHR2. The authors reported that both the gene-environment interaction, i.e. FKBP5–environment interaction, and gene-environment interaction, i.e. FKBP5–CRHR1 interactions affected PTSD [116]. Zhang and colleagues showed that FKBP5 influenced different symptoms of the stress response in diverse ways, which revealing the complex genetic architecture of the HPA axis genes in response to stress [116].

Next, we assessed AVP; while AVP is known to play a role in propagating brain neurotoxicity in traumatic brain injury and cerebral edema [117,118,119], increased levels of AVP have also been reported to play a primary role in the regulation of the HPA axis during adaptation to stress [120,121,122]. Since the main role of AVP within the HPA axis is the potentiation of the effects of CRH on the release of ACTH [123,124,125], and since it is thought to be secreted in a pulsatile pattern, it was theorized that increased expression of AVP may potentiate the pulsatile activity of the HPA axis [125,126]. It was later found that AVP does not regulate basal HPA axis activity [127,128] and has no effect on basal circadian and ultradian rhythms of corticosterone secretion [129]. 

The involvement of the vasopressinergic system in the PSS-induced behavioral and HPA axis disruptions was evaluated 8 days after exposure by examining the levels of AVP-positive cells and fibers in the PVN and SON hypothalamus nuclei. We found that the PTSD-phenotype was associated with a pattern of persistent and excessive expression of AVP in the PVN and SON nuclei, the sites in which AVP gene expression and synthesis are strongly affected by glucocorticoids [130,131,132]. Our present results are in good agreement with previous clinical findings and preclinical rodent studies. PTSD patients were reported to have significantly elevated plasma AVP levels com-pared to healthy controls [133,134,135], although strong contradictory findings exist [136,137]. 

Surprisingly, over-expression of AVP in the PTSD-phenotype group was evident in both the magno- and parvo-cellular cell groups; these are two separate neurosecretory systems, with different physiological functions, morphology, origins, electrophysiology, and projections [138,139,140]. This finding, in combination with current literary evidence of the lack of effect of AVP on corticosterone pulsatility and contradictory findings of AVP levels in PTSD, might suggest the changes in AVP are a downstream consequence of the original fault in regulation and not a vulnerability trait.

It is important to emphasize that the dysregulation in basal and reactive HPA axis in PTSD is complex and may also involve a variety of steroid compounds, including the neuroactive steroids, such as dehydroepiandrosterone (DHEA), its sulphate ester dehydroepiandrosterone sulphate (DHEA-S), allopregnanolone and pregnanolone. Alterations in the synthesis of neuroactive steroids has been associated with increased PTSD symptoms in humans [141,142,143] and in animal studies [144,145,146,147]. A marked decrease in cerebrospinal (CSF) levels of allopregnanolone and its equipotent stereoisomer, pregnanolone has been found in women with PTSD, and is apparently due to a block in ALLO synthesis at 3α-HSD [141]. These data in humans align with animal models studies [145,148,149]. A significantly decreased DHEA levels were found 7 days post-exposure only in PTSD-phenotype group, not in their MBR-phenotype counterparts [144] Moreover, DHEA-S levels were reduced in both EBR and MBR PSS-exposed rats compared to unexposed controls [144]. These results suggests that concomitantly decreased circulatory levels of DHEA and elevated corticosterone levels may be associated with an extreme (pathological) response to stress, whilst maintenance of normal levels of both steroids may be associated with minimal response, denoting resilience [144]. 

Our study is not devoted of limitations in establishing the exact interaction, this owing to measuring the regulating factors at different time points, due to the nature of brain immunohistochemistry technique. Another limitation is that the PSS-stress response was not measured thus we can only correlate pre-trauma suppression patterns to behavioral outcomes and not the stress reaction itself, furthermore serial blood samples during MPST were not taken for a long enough period to establish a clear pulsatile pattern. An additional limitation of this study is the relatively small sample size of the MPST experiment (*n* = 21) resulting in a small number of EBR rats (*n* = 4 Larger samples would have been preferable in the other experiments presented in this study, although the overall sample size was sufficient for statistical analyses. These limitations highlight the need for further studies into the involvement of HPA-axis reactivity (i.e., positive and negative feedback) and its regulating factors as predictors and/or origins of PTSD development.

In conclusion: The data presented indicate a sensitized response to methylprednisolone challenge, associated with a more robust corticosterone pulsatile activity, by EBR rats (PTSD-Phenotype group) in comparison with MBR (resilient group) rats. Moreover, they suggest a role for increased GR and MR availability in the dorsal hippocampus and reduced MR and GR availability in the ventral hippocampus in the development of PTSD-like behavioral responses by stressed rats (in both cases, the effects of stress in EBR are significantly different from the effects in MBR but not significantly different from the effects in PBR). These findings fit well with previous results obtained by our research group and support the value of their PTSD model. The most significant findings of this study are: (1) Over-sensitivity of the HPA axis, as evident by hyper-suppression of the HPA axis in response to MPST, suggests a pre-trauma vulnerability factor for traumatic exposure, which may predict the subsequent development of PTSD. (2) This pre-trauma vulnerability is characterized not only by enhanced feedback inhibition of the HPA axis but by an overall over-regulation of this axis, including an over-reactiveness of forward-driving mechanisms as seen through the prism of the increased size and number of amplitudes in conjunction with hyper-suppression during an MPST. (3) PTSD phenotype rats exhibited long-term regionally distinct alteration of the signaling of GR-MR-FKBP5 in the hippocampus and in the hypothalamic nucleus in response to PSS, attenuated GR and MR, complemented by elevation in FKBP5, expression in the ventral hippocampus and PVN, and augmented GR and MR complimented by a decrease in the FKBP5 expression in the dorsal hippocampus. This is important for glucocorticoid feedback inhibition of the HPA axis and stress signaling cascade, as well as other mechanisms of fear extinction and memory consolidation. (4) PTSD phenotype rats exhibited specific changes in the patterns of AVP expression in the hypothalamic nuclei PVN and SON. (5) Not all exposed rats subsequently developed PTSD phenotype, and not all pre-existing vulnerability factors/risk factors determined the individual’s likelihood of developing PTSD phenotype upon stress exposure. (6) The association between risk factors and diseases is complex and multifactorial, influenced by a variety of factors involving genetic risk and environmental conditions interfacing with a variety of potentially interacting “hits” and vulnerability factors occurring during exposure, contrary to the binary “two hit model” that was proposed nearly 30 years ago.

## 4. Materials and Methods

All procedures were performed under strict compliance with ethical principles and guidelines of the NIH Guide for the Care and Use of Laboratory Animals. All treatment and testing procedures were approved by the Animal Care Committee of the Ben-Gurion University of the Negev, Israel 18 December 2018 (IL-06-09-2018).

### 4.1. Animals

A total of 30 adult male Sprague-Dawley rats, weighing 190–320 g (21 rats underwent MPST and 9 rats for control), were used in this study. The rats were housed (three per cage) in a vivarium with a stable temperature under a 12:12-h light-dark cycle (lights off at 19:00; luminous emittance during the light phase: 200G50 lux), with unlimited access to food and water. All rats were allowed a 1-week habituation period before the experiment. All procedures were performed during the resting phase of the rats between 08:30 and 12:00. After jugular vein catheter surgery, rats were housed singly in opaque plastic bins with restricted access to one another through a mesh barrier. The mesh barrier cages consist of a 40 cm × 28 cm × 20 cm high transparent Perspex box Each chamber contained a removable partition that separated the chamber into two equal-sized compartments. The mesh barrier was made of wire grid mesh (1.5 cm × 1.5 cm), allowing the rats to insert their noses into the adjacent compartment. Rats were handled daily following jugular catheterization.

### 4.2. Experimental Design

This study aimed at assessing the strength of negative feedback sensitivity of the HPA axis and regulating factors. We used methylprednisolone suppression tests (MPSTs), which were performed 7 days following the post-surgical recovery period. Serial blood samples were collected manually via a jugular vein cannula in conscious rats at 10-min intervals to evaluate the suppression of corticosterone following methylprednisolone (sodium succinate, Solu-Medrone; Pharmacia & Upjohn Ltd., Milton Keynes, UK) (945 µg/kg). After 24 h, rats were exposed to PSS or sham-PSS. Behaviors were assessed on day 7 using the EPM and the ASR paradigms. The prevalence rates of extreme, partial, and minimal behavioral responses (EBR, PBR, and MBR) were assessed. One day later, rats were sacrificed, and their brains were collected for an immunoreactivity analysis of GR-ir, MR-ir, AVP-ir, and FKBP52-ir in hippocampus subregions and hypothalamic nuclei.

Our study focused on brain regions essential for glucocorticoid feedback inhibition of the HPA axis and stress signaling cascade, including the PVN and dorsal and ventral hippocampus. The hypothalamic nuclei and hippocampus were chosen as targets in this study because these two regions appear to be key sites for glucocorticoid feedback inhibition of the HPA axis [27,28,85]. While the paraventricular nucleus (PVN) is essential for appropriate initiation and termination of the stress response [29], the hippocampus is thought to be one of the most important feedback sites in the brain for the activity within the HPA axis [150]. In addition, the hippocampus has also been implicated in PTSD neurobiology because it has been demonstrated to play an important role in fear learning, extinction, and conditioning/acquisition of traumatic memories [151,152,153,154,155,156]. Moreover, the dorsal (DH) and ventral (VH) regions of the hippocampus are anatomically and functionally distinct areas [76]. The DH, which is connected mainly through differential connections with the neocortex, performs primarily cognitive functions, whereas the VH (connected mainly with the amygdala and hypothalamus) is associated with stress and motivational and emotional behaviors [157,158,159]. Therefore, we examined GR, MR, and FKBP5 protein levels in the dorsal and ventral hippocampus separately. In the rat, there are three hypothalamic nuclei that contain large numbers of AVP cells: the anterior commissural nucleus, the PVN, and the supraoptic nucleus (SON). In this study, we evaluated the density of AVP in the PVN and SON. The experimental design used here is schematically depicted in Figure 1.

### 4.3. Predator Scent Stress (PSS)

Rats were individually placed on well-soiled cat litter, which was used by a cat for 2 days and sifted for stools. The rats were exposed to the litter for 10 min in a plastic cage (inescapable exposure) placed on a yard paving stone in a closed environment [20,21,22,23,160]. Sham-PSS (control) was administered under similar conditions, but the rats were exposed to a fresh, unused cat litter.

### 4.4. Behavioral Measurements

The behavior of rats was assessed in the EPM and ASR paradigms, as described previously [20,21,22,23,160] and as briefly detailed in Appendix A. All behavioral tests were video-recorded for future analysis using the ETHO-VISION program (Noldus Information Technology, Wageningen, The Netherlands) by an investigator blinded to the experimental protocol.

### 4.5. Cut-Off Behavioral Criteria (CBC) Model

The classification of individuals according to the degree to which their behavior was affected by a stressor is based on the premise that extremely compromised behavior in response to the priming trigger is not conducive to survival and is thus inadequate and maladaptive at representing a pathological degree of response. The severity of the response of each rat to the stressor is classified as either an extreme behavioral response (EBR), a partial behavioral response (PBR), or a minimal behavioral response (MBR) [20,21,22,23,160] (See Appendix A). 

### 4.6. Mesh Barrier Housing

All rats that underwent jugular catheterization surgery had to be isolated to prevent tampering with the cannula and surgical stitches. To prevent isolation anxiety, rats were handled daily and put in housing that was separated by a mesh barrier as described elsewhere (see Klapper-Goldstein et al. Appendix A [161]).

### 4.7. Jugular Vein Catheter Surgery and Blood Sampling

Rats were anesthetized with a combination of ketamine (70 mg/kg) and xylazine (6 mg/kg) i.p. The right jugular vein was exposed, and a polyurethane cannula (inner diameter 0.63 mm, outer diameter 1.02 mm, Instech, Plymouth Meeting, PA, USA) was inserted into the vessel until it lay close to the entrance of the right atrium. The cannula was prefilled with pyrogen-free heparinized (10 IU/mL) isotonic saline. The free end of the cannula was exteriorized through an incision between the shoulder blades and then capped. Following surgery, animals were housed in individual cages for recovery for 7 days prior to beginning the blood sampling protocol. Jugular cannulas were manually flushed daily by withdrawing 0.2 mL of blood and replacing the volume with 0.3 mL of sterile heparinized saline (0.2 mL to replace blood volume; 0.1 mL to clear blood from cannula) to maintain patency of the cannula. For the blood sampling following recovery, the end of the capped cannula was attached to a special harness connected to a mechanical swivel via a long metal spring that rotated 360° in a horizontal plane and 180° through a vertical plane to maximize freedom of movement for the rats (Cat. No. 375/22, Instech Laboratories). For Experiment 1, blood samples were collected by hand (700 µL) every 20 min (for a duration of 11.5 h); for Experiment 2, blood samples were collected by hand (300 µL) every 10 min (for a duration of 90 min). Blood samples were stored in Eppendorf tubes. Serum was separated by centrifugation and then stored at −80 °C until processed for corticosterone measurements.

### 4.8. Measurement of Serum Corticosterone

Corticosterone was measured with a DSL-10-81000 ELISA kit (Diagnostic Systems Laboratories, Webster, TX, USA) according to the instructions of the manufacturer by a person blind to the experimental procedures. All samples were measured in duplicates.

### 4.9. Methylprednisolone Suppression Test

A methylprednisolone suppression test was used to measure feedback inhibition of the HPA axis activity. Each rat received a push intravenous injection of 165 μg/kg (approx. 40–50 μg) methylprednisolone (sodium succinate, Solu-Medrone; Pharmacia & Upjohn Ltd., Milton Keynes, UK), and blood samples were collected at baseline (prior to methylprednisolone injection), and every 10 min for a duration of 90 min starting at 40 min post-methylprednisolone administration. Due to a prominent yet more rapid onset of GC suppression and faster recovery, methylprednisolone was chosen over dexamethasone for the GC suppression test, as it better fits our technical limitations of serial blood sampling [26,162,163]. Dose and time of sampling were based on a previous study by Andrews et al. [164].

### 4.10. Immunohistochemistry

Tissue preparation: Twenty-four hours after the behavioral tests, rats were deeply anesthetized and perfused transcardially with cold 0.9% physiological saline, followed by 4% paraformaldehyde (Sigma-Aldrich, St. Louis, MI, USA) in a 0.1 M phosphate buffer. Brains were quickly removed, postfixed in the same fixative for 12 h at 4 °C, cryoprotected overnight (30% sucrose in 0.1 M phosphate buffer at 4 °C), and then frozen and stored at −80 °C. Serial coronal sections (10 μm) were obtained with a cryostat (Leica CM 1850; Leica Microsystems, Wetzlar, Germany) and mounted on coated slides.

Staining: Sections were air-dried and washed three times in phosphate-buffered saline (PBS) containing Tween 20 (PBS/T) (Sigma-Aldrich, St. Louis, MI, USA). They were incubated for 60 min in a blocking solution (normal goat serum in PBS) and then overnight at 4 °C with the primary antibodies against GR [rabbit polyclonal anti-GR antiserum (1:300), product code: sc-ANT-010, Alomone Labs, Jerusalem, Israel), MR (rabbit polyclonal anti-MR antiserum (1:300), product code: sc-ANT-010, Alomone Labs, Jerusalem, Israel), AVP (Rabbit Anti-Vasopressin antibody (1:500), product code: ab68669, Abcam, Israel), FKBP5 (rabbit polyclonal anti-FKBP5 antiserum (1:500), product code: ab2926, Abcam, Israel). After three washes in PBS/T, the sections were incubated for 2 h in DyLight-488-labeled goat-anti-rabbit IgG or in Dylight-594 goat-anti-mouse IgG (1:250; KPL, Gaithersburg, MD) in PBS containing 2% normal goat or horse serum. The sections were subsequently washed and mounted with a mounting medium (Vectastain; Vector Laboratories, Burlingame, CA, USA). Sections of the brains of different groups of rats were processed at the same time and under identical conditions to ensure reliable comparisons and to maintain stringency in tissue preparation and staining conditions. Control staining was performed in the absence of the primary antibodies. Additionally, secondary fluorescent labels were swapped to test for cross-reactivity, and sections were incubated without primary antibodies to test for non-specific binding of the secondary antibodies.

AVP immunoreactivity was examined in parvo- and magnocellular parts of the hypothalamic PVN and in the SON. Magnocellular neurons were differentiated histologically from the parvocellular neurons based on their overall size, morphology, and location, as described in previous studies of the PVN cellular mappings [164,165].

### 4.11. Quantification of MR and GR Translocation Patterns in Hippocampal Subregions

To quantify differences in MR and GR subcellular distribution patterns, changes in fluorescence intensity values of nuclear immunoreactivity were measured, using ImageJ 1.32j analysis software (NIH, Stapleton, NY, USA; http://rsb.info.nih.gov/ij/, accessed on 3 March 2021), in the same manner as described previously [166,167,168]. Briefly, Hoechst staining was used to identify the boundaries between the nuclear surface and cytoplasm of individual cells and was circled with the analysis software. These circles served as a template and were pasted onto the corresponding MR and GR images to measure the optical density (mean grey levels) within the nucleus. Non-specific binding (normal mouse and rabbit IgG) and background staining of the sections were also measured and subtracted from the total signal to obtain the specific signal. Only cells that had a clear oval-shaped nucleus with a diameter of approximately 5–7 μm and showed IR clearly above the background were included for analysis, thereby excluding cells that were not in the plane of focus.

### 4.12. Statistical Analyses

Data are presented as the mean ± SEM unless otherwise specified. *p* < 0.05 was statistically significant. For the behavioral results and for serum corticosterone levels, the statistical analyses were performed using RM-ANOVA. Bonferroni tests were used to examine differences between individual groups. The behavioral data were transformed to a percentage by using the CBC model: the prevalence of affected rats as a function of the rat group was tested by using cross-tabulation and nonparametric Chi-squared tests. All nonparametric analyses were performed on raw data (and not on percentage). Corticosterone pulses analysis: All corticosterone ultradian pulsatile data were analyzed using an adaptation of Hierarchically AdaPtive analysis, an algorithm written as a Matlab-based code, designed for analysis of hormonal pulsatility, created by Dean et al. [25]. The amplitudes were determined as the difference in corticosterone levels between each nadir and its subsequent peak; the algorithm was unable to identify interpulse intervals as they were determined as the time in minutes between two adjacent peaks.

For the GR, MR, and FKBP5 levels, statistical analyses were performed with a two-way ANOVA with groups (EBR, PBR, and MBR vs. control) and hippocampus axis (dorsal and ventral poles of the hippocampus) as the independent factors, or one-way ANOVA (for the PVN area). For the AVP levels, statistical analyses were performed with a one-way ANOVA. Bonferroni tests were used to examine differences between individual groups. To gain an additional understanding of the relationship between behavioral measures and corticosterone pulsatility parameters, two sets of regression analyses were conducted. Pearson’s correlation analysis was used to describe the relationship between each behavioral variable (time spent in the open arms, startle amplitude, and startle habituation) and corticosterone pulsatility parameters across the entire sample. The contribution of each variable that exhibited a significant correlation (or trend) was then evaluated using a multiple-stepwise regression analysis. An alpha level of *p* < 0.05 was considered statistically significant.

## Figures and Tables

**Figure 1 ijms-22-06050-f001:**
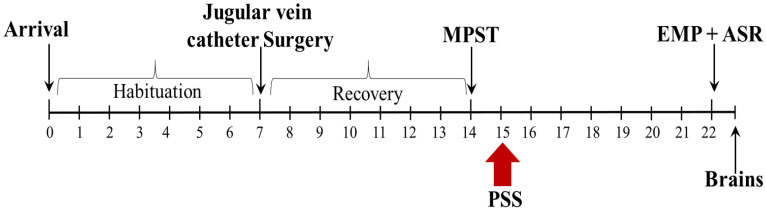
The timeline of the Methylprednisolone Suppression Test (MPST) experiment. Rats were habituated, and their jugular veins cannulated as described (see Methods). The corticosterone level was measured for baseline (BL), and 10 min later, methylprednisolone was administered through the cannula. Serial blood samples were collected starting at 20 min post-methylprednisolone administration at 10-min intervals for 90 min to evaluate the suppression of corticosterone following methylprednisolone administration. One day later, rats were exposed to PSS, and behavioral assessments were conducted 7 days post-exposure to PSS, first in the EPM test and 1 h later in the ASR test. On the following day, the rats were sacrificed, and their brains were collected and frozen for future immunohistochemistry assays.

**Figure 2 ijms-22-06050-f002:**
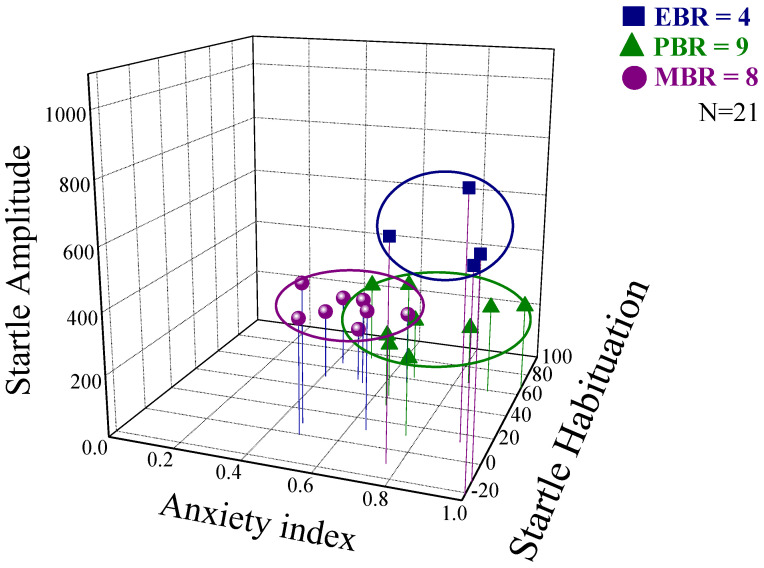
Behavioral responses to PSS observed in the MPST experiment. Rats were exposed to PSS. Their behavior was assessed 7 days later in an elevated plus maze (EPM) test and then one hour later in an acoustic startle response (ASR) test. Data are displayed in three dimensions: The X-axis represents time spent in the open arms (min) in the EPM test. The Y-axis represents acoustic startle amplitude in the ASR test, and the amplitude of the whole-body startle to an acoustic pulse was defined as the average of 100 one-millisecond accelerometer readings collected from pulse onset. The Z-axis represents startle habituation (percentage) in the ASR test. Rats were grouped by their behavioral responses according to the cutoff behavioral criteria (CBC) model (see Methods) as having either an extreme behavioral response (EBR) to PSS (i.e., PTSD-phenotype), a partial behavioral response (PBR) or a minimal behavioral response (MBR). Squares represent the EBR group that exhibited a significant degree of anxiety-like and avoidant behaviors on the elevated plus-maze and a pattern of exaggerated startle responses with significantly reduced habituation 7 days after PSS exposure. Squares represent the EBR group; Triangles represent the PBR group; Circles represent the MBR group [20,21].

**Figure 3 ijms-22-06050-f003:**
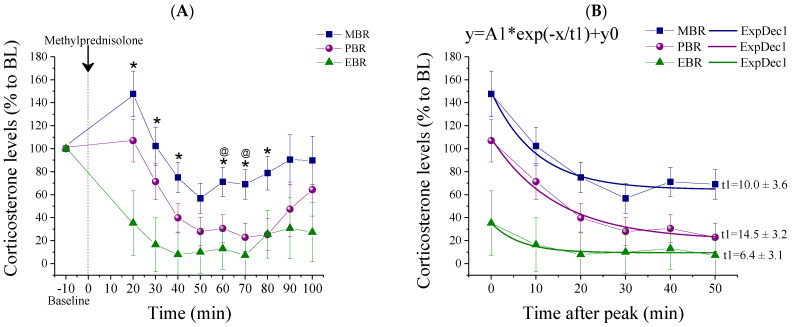
Corticosterone measurements post-MPST. Corticosterone measurements following MPST depicted as percentages from the baseline between the different CBC groups. (**A**) Measurements starting at 10 min before methylprednisolone injection (baseline) and 20 min after methylprednisolone injection, at 10 min intervals over a period of 90 min, points are averaged within each group (* MBR ≠ EBR, *p* < 0.04; @ MBR ≠ PBR *p* < 0.05; mean ± SEM). (**B**) A nonlinear fitting decay model carried out on the 6 time points following corticosterone peak concentrations indicated significantly different decay constants (*t*1) between groups, such that EBR rats exhibited a slower decay in corticosterone levels than MBR and PBR rats.

**Figure 4 ijms-22-06050-f004:**
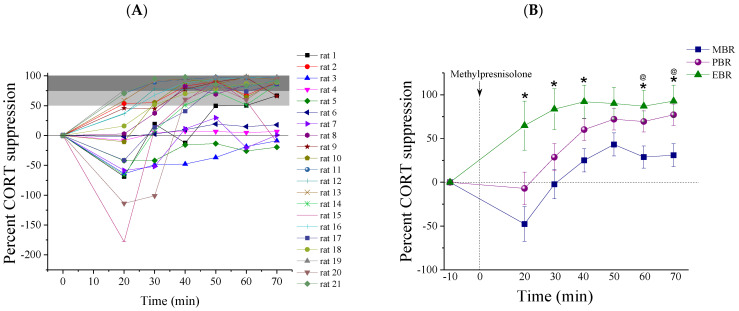
The percent of corticosterone suppression post-MPST. Corticosterone suppression in 10 min intervals over 60 min following MPST, depicted as a percentage of suppression from the baseline. (**A**) The percentage of suppression plotted for all individual rats. (**B**) The percentage of suppression between the different CBC groups (* MBR ≠ EBR, *p* < 0.04; @ MBR ≠ PBR *p* < 0.05; mean ± SEM).

**Figure 5 ijms-22-06050-f005:**
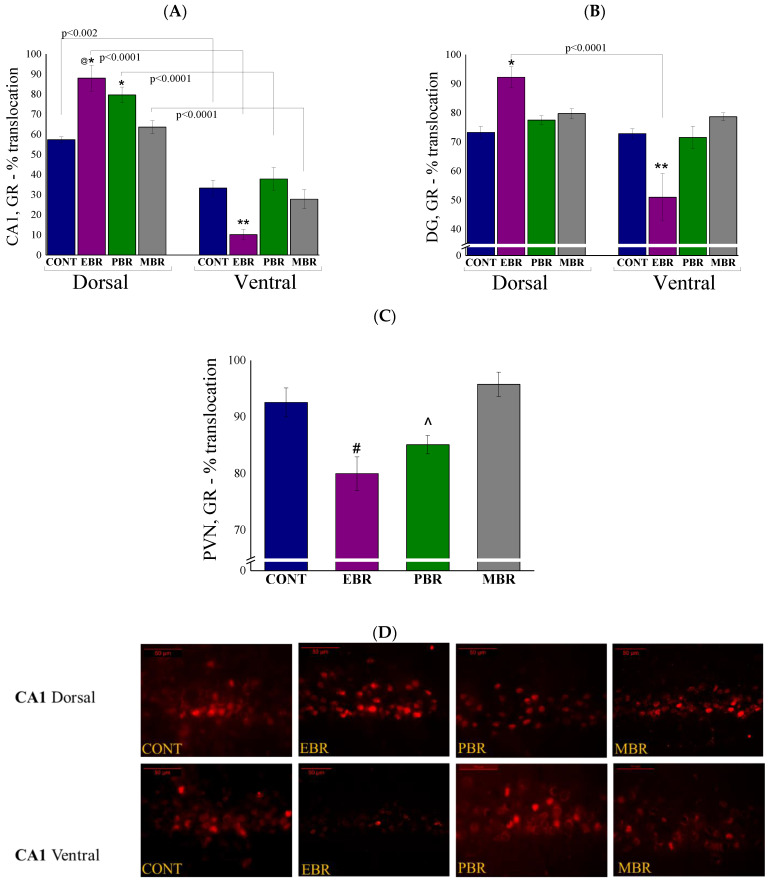
Brain GR-ir cells 8 days post-PSS exposure. Immunohistochemistry of GR in specific brain areas 8 days post-PSS exposure. Significant differences between dorsal and ventral parts are depicted in the graphs, whereas notations are given for significant differences between groups in the same area and are explained in this legend for each graph separately. (**A**) GR-IR in the CA1 region of the ventral and dorsal hippocampus [* significant as compared to dorsal control (*p* < 0.002); @ significant as compared to dorsal MBR (*p* < 0.04); ** significant as compared to ventral control and PBR (*p* < 0.05)]. (**B**) GR-IR in the ventral and dorsal dentate gyrus [* significant as compared to dorsal control (*p* < 0.04); ** significant as compared to ventral control, PBR and MBR (*p* < 0.009)]. (**C**) GR-IR in the PVN [# significant as compared to control and MBR (*p* < 0.02); ^ significant as compared to MBR (*p* < 0.015)]. (**D**) Representative images of GR-immunoreactivity in the dorsal and ventral CA1 subregion of CONT, EBR, PBR, and MBR groups. Images were acquired at 40× magnification. Scale bar: 50 μm. The cells in red are GR-positive.

**Figure 6 ijms-22-06050-f006:**
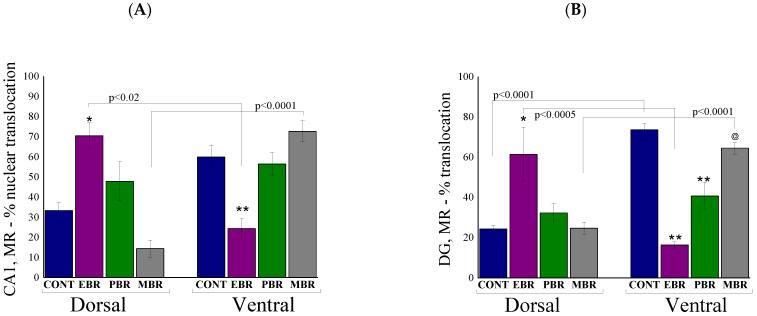
Brain MR-ir cells 8 days post-PSS exposure. Immunohistochemistry of MR in specific brain areas 8 days post-PSS exposure. Significant differences between dorsal and ventral parts are depicted in the graphs, whereas notations are given for significant differences between groups in the same area and explained in this legend for each graph separately. (**A**) MR-IR in the CA1 region of the ventral and dorsal hippocampus [* significant as compared to dorsal control and MBR (*p* < 0.03); ** significant as compared to ventral control and MBR (*p* < 0.045)]. (**B**) MR-IR in the ventral and dorsal dentate gyrus [* significant as compared to dorsal control, PBR and MBR (*p* < 0.02); ** significant as compared to ventral control (*p* < 0.001); @ significant as compared to ventral EBR and PBR (*p* < 0.02)]. (**C**) MR-IR in the PVN [* significant compared to control (*p* < 0.015)]. Panel (**D**) shows a schematic drawing of the regions of the PVN from which the measurements were collected. The image was acquired at 10× magnification. Scale bar: 200 μm. Additionally, in (**E**), there are representative images of MR-immunoreactivity in the PVN region of CONT, EBR, PBR, and MBR groups. Images were acquired at 20× magnification. Scale bar: 100 μm. The cells in red are MR-positive.

**Figure 7 ijms-22-06050-f007:**
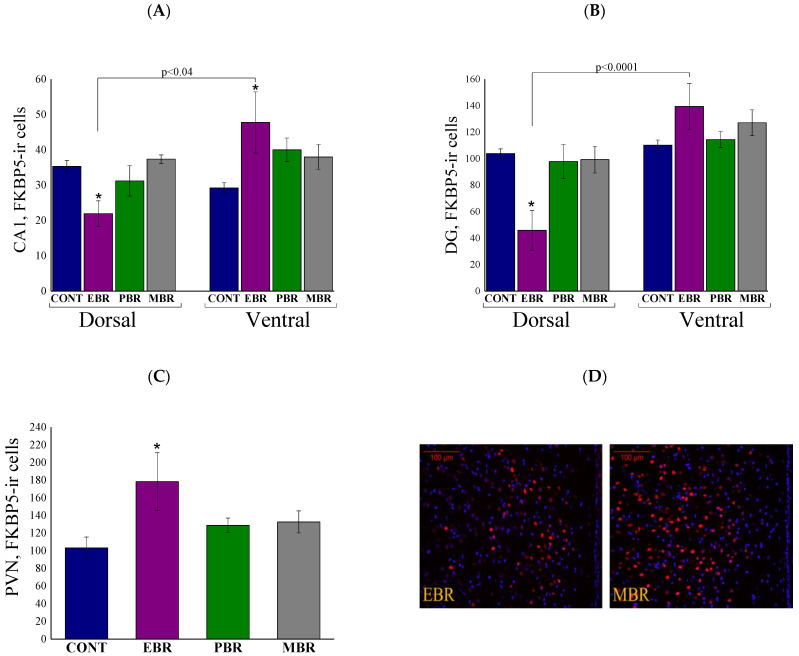
Brain FKBP5 levels 8 days post-PSS exposure. Immunohistochemistry of FKBP5 levels in specific brain areas 8 days post-PSS exposure. Significant differences between dorsal and ventral parts are depicted in the graphs, whereas notations are given for significant differences between groups in the same area and explained in this legend for each graph separately. (**A**) FKBP5 in the CA1 region of the ventral and dorsal hippocampus [* significant as compared to control (*p* < 0.05)]. (**B**) FKBP5 in the ventral and dorsal dentate gyrus [* significant as compared to dorsal control (*p* < 0.015)]. (**C**) FKBP5 in the PVN [* significant as compared to control (*p* < 0.0025)]. (**D**) Representative images of FKBP5-immunoreactivity in the PVN region of EBR and MBR groups. Images were acquired at 20× magnification. Scale bar: 100 μm. The cells in red are FKBP5-positive. (**E**) Representative images of FKBP5-immunoreactivity in the DG region of EBR and MBR groups. Images were acquired at 40× magnification. Scale bar: 50 μm. The cells in red are FKBP5-positive.

**Figure 8 ijms-22-06050-f008:**
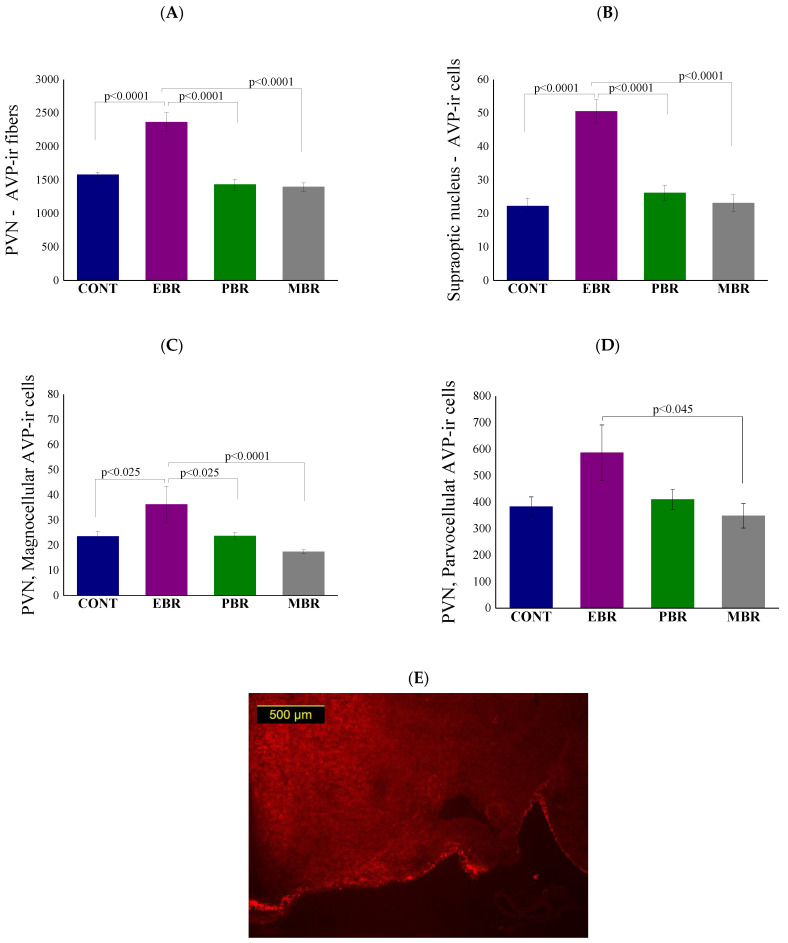
Brain AVP levels 8 days post-PSS exposure. Immunohistochemistry of AVP 8 days post-PSS exposure. (**A**) Levels of AVP in overall fibers in the PVN. (**B**) Levels of AVP in the overall cells of the Supraoptic nucleus. (**C**) Levels of AVP in magnocellular neurosecretory cells in the PVN. (**D**) Levels of AVP in magnocellular neurosecretory cells in the PVN. Significant values are shown in the graphs. Panel (**E**) shows a schematic drawing of the region of the SON from which the measurements were collected. (**F**) Representative images of AVP-immunoreactivity in the SON region of CONT, EBR, PBR, and MBR groups are also shown. Images were acquired at 10× magnification. Scale bar: 200 μm. The cells in red are MR-positive.

**Table 1 ijms-22-06050-t001:** The percentage of corticosterone suppression at 20, 40 and 70 min after methylprednisolone administration.

Time Post Methylprednisolone Administration	EBR *n* = 4	PBR *n* = 9	MBR *n* = 8	Statistics
20 min				
0–50% Suppression, (*n*)	0	100%, (9/9)	87.5%, (7/8)	Pearson χ^2^ = 12.35, df = 2, *p* < 0.0025
51–89% Suppression, (*n*)	100%, (4/4)	0	12.5%, (1/8)	
90%+ Suppression, (*n*)	0	0	0	
40 min				
0–50% Suppression, (*n*)	0	22%, (2/9)	75%, (6/8)	Pearson χ^2^ = 11.23, df = 4, *p* < 0.025
51–89% Suppression, (*n*)	25%, (1/4)	67%, (6/9)	12.5%, (1/8)	
90%+ Suppression, (*n*)	75%, (3/4)	11%, (1/9)	12.5%, (1/8)	
70 min				
0–50% Suppression, (*n*)	0	11%, (1/9)	62.5%, (5/8)	Pearson χ^2^ = 11.12, df = 4, *p* < 0.03
51–89% Suppression, (*n*)	0	33%, (3/9)	25%, (2/8)	
90%+ Suppression, (*n*)	100%, (4/4)	56%, (5.9)	12.5%, (1/8)	

**Table 2 ijms-22-06050-t002:** Average pulsatility during MPST suppression and recovery. Data represent group mean ± SEM.

Average Pulsatility during MPST	EBR *n* = 4 (I)	PBR *n* = 9 (II)	MBR *n* = 8 (III)	Statistics
Amplitudes average suppression ^1^	0.16 ± 0.1	0.036 ± 0.02	0	I ≠ III (Bonferroni) I ≠ II, III (Fisher LSD)
Amplitudes average recovery ^2^	0.64 ± 0.15	0.53 ± 0.13	0.24 ± 0.09	NS

^1^ F(2, 18) = 4.1213, *p* < 0.035. ^2^ F(2, 18) = 2.5073, *p* = 0.11.

## Data Availability

Not applicable.

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
