# Peer review of "Is PTSD-Phenotype Associated with HPA-Axis Sensitivity? Feedback Inhibition and Other Modulating Factors of Glucocorticoid Signaling Dynamics"

_ijms, 2021, doi:10.3390/ijms22116050_

Round 1

Reviewer 1 Report

Interesting research on the relationship of the HPA axis and PTSD and the dynamics of glucocorticoid signalling. However, I have a few reservations: 
Firstly, the title: the authors used the word part 1. I expected, after such a headline, some kind of announcement of further studies, which were not included if only because of the volume of the publication. Meanwhile, nothing of the sort took place. 
Please change the title, as it is misleading
Results: 
Immunohistochemistry was the main method used to determine individual levels of GR, MR, AVP and FKBP52 . Why are there no images attached? Please incude images for all parameters determined for each group and individual structures. 

Reviewer 2 Report

This study, in line with a number of previous ones by the same research group, is well conducted and focuses on neuroendocrine and behavioral phenotypes with relevant translational value.

My main concern with the reported data is the extremely low number of rats in the experimental group (EBR=4). Some of the findings are not so strong and the reader wanders whether a larger experimental group could have produced different results.

A second, although not secondary, problem is in the discussion of the findings

  1. Sometime statements are in clear conflict with data. As an example, in the middle paragraph of page 15, authors state:             ” Moreover, we found that the PVN, anatomically connected to the ventral hippocampus, shows a similar GR-MR outcome, i.e., significantly decreased GR and MR in PTSD-phenotype rats.” Results obtained with GR and MR measures are not the same and this is especially true for PVN. In the PVN of EBR rats (rats expressing the PTSD phenotype) GR are significantly reduced not only in comparison with unstressed rats but also in comparison with MBR (resilient rats); whereas MR are equally reduced in all the stressed rats regardless of the expressed phenotype. Because the behavioral phenotype is the only differences between stressed groups the conclusion that should be derived from these data is that the effects of stress on PVN MR do not seem to be related with expression of the pathological phenotype. Moreover, there is clearly no relationship between the effects of stress on PVN and hippocampal MR.
  2. In the same paragraph, few lines below, we find a series of rather confusing statements discussing human findings on “ … loss of resting-state functional connectivity between the dorsal hippocampus and default mode network areas..”  in PTDS patients and “ Segal et al. [59] proposed that acute stress could induce a disconnect between the dorsal and ventral hippocampus..” that should indicate a homology between the observed differences for the measures collected  in the two areas of the hippocampus (ventral and dorsal) by the present study. However, significant differences between data collected in the dorsal and ventral hippocampus were also evident in the unstressed group.
  3. “Conclusions: this work has shown that there are several key elements in the neuroendocrine system that modulate the pre-stress, during stress and post-stress biological and behavioral responses; these elements affect each other and interact across distinct separate neurological and endocrine systems and can increase resilience or susceptibility to stress and subsequent symptoms. “Unfortunately, I missed the modulatory and interactive experiments, they would have been interesting. The data presented indicate a sensitized response to methylprednisolone challenge, associated with a more robust corticosterone pulsatile activity, by EBR rats (which develop PTSD-like behavioral phenotypes following stress) in comparison with MBR (stress resilient) rats. Moreover, they suggest a role for increased GR availability in the dorsal hippocampus and reduced MR availability in the ventral hippocampus in the development of PTSD-like behavioral phenotypes by stressed rats (in both cases the effects of stress in EBR are significantly different from the effects in MBR but not significantly different from the effects in PBR). These finding fit well with previous results obtained by this research group and support the value of their PTSD model.

Round 2

Reviewer 1 Report

Thank you for responding to the review.

Author Response

We added more information to the discussion section regarding the HPA axis and neuroactive steroids as recommended. 
